# Filling Process Optimization through Modifications in Machine Settings

**Yanmei Cui [1], Xupeng Zhang [2],\*  and Jing Luo [3]**

[1]  School of Mechanical Engineering, Shanghai Dianji University, Shanghai 201306, China
[2]  Research and Development Department, Nexon Electronic Co., Ltd., Shanghai 201107, China
[3]  The School of Engineering Science, University of Chinese Academy of Sciences, Beijing 100864, China
\*  Correspondence: zhangxupeng0905@126.com or zhangxupeng1@nexonelectronic.cn

**Abstract:** In this paper, a mathematical model is developed for the modified settings of an automatic filling machine to minimize the filling time of orders for different volumes of dairy product and flavors. The linear programming model is solved using the Simplex method to find an optimal solution to the optimization problem. The results of the model are used for sequencing the processing of orders using one-dimensional rules with the aim of obtaining an optimal sequence for the most valued performance measure. The comparative analysis of the one-dimensional rules showed that Shortest Processing Time (SPT) is better than the other rules for minimization of the average time past due. Additionally, the results of the model for the new machine settings, when compared with previous similar studies, yielded encouraging results.

**Keywords:** machine scheduling; bottle filling; one-dimensional rules; process optimization; mathematical modeling



## 1. Introduction

The orders that are released in a manufacturing environment are converted into jobs by following predetermined scheduling, processing time, machining routing, and priority rules. There can be a delay in the processing of jobs when certain machines are busy. Preemption may also occur in case of the arrival of high-priority jobs for processing. Longer-than-expected processing delays and equipment failures are further unanticipated situations that might be considered. In these situations, developing a comprehensive schedule aids in supporting operational control and productivity. Hence, manufacturing scheduling is considered an important decision-making process for production systems [1]. Scheduling problems in different manufacturing environments may vary from single-machine scheduling to parallel machine scheduling. In single-machine scheduling, several jobs are undertaken with different processing times scheduled on a single facility such that the throughput is optimized. In parallel machine scheduling, multiple facilities of similar type are used, and the jobs are scheduled according to the available machines to maximize the throughput. However, examples of unconnected parallel machines have received less attention, particularly when setup times are considered [2–6]. Due to its importance to the manufacturing environment, parallel machine programming was thoroughly explored in the early 20th century [7,8]. Manufacturing scheduling is important since it directly or indirectly optimizes the system's configurations and productivity [9–15].

The required volume of a liquid can be filled into the bottle by adjusting the height of the liquid in the bottle. The bottles are used as measuring vessels by the level-controlled fillers [16]. The volumetric filling systems fill a known volume of a liquid into a bottle. These systems divide the liquid into required volumes before filling it into the bottle. The liquid is poured from the supply tank into the measuring glass and the required quantity of a liquid is then filled into the bottle [17]. P. Birmole et al. [18] worked on the filling of bottles using a programmable logic controller (PLC). The aim of the research

work was the development of PLC-based automatic mixing of colors and filling of bottles. The coordination of supervisory control and data acquisition (SCADA) and PLC was presented in detail. Additionally, the benefits of automation in industries were discussed. F. Basso et al. [19] worked on the bottle scheduling problem in the wine industry. The main contribution of the research work was the development of a novel mathematical model for the bottling process. It was shown that the suggested model can find a better solution than CPLEX. A. N. Abubakar et al. [20] worked on the automated liquid filling system. The overall system was developed using an Arduino controller and a robotic arm. For the management of the movement of the robotic arm and sensing the presence of bottles, gear motors and infrared (IR) proximity sensors were used, respectively. The working of the system was found satisfactory when tested under various conditions. F. Basso et al. [21] worked on the bottling scheduling problem. Their contribution includes the achievement of considerable saving through numerical experiments. K.S. Kiangala et al. [22] developed an auto parameter configuration of a bottling process. The proposed strategy allowed the industrial supervisors to be free from the configuration of manual bottling process parameters and to directly monitor the production steps.

The dispatching rules or one-dimensional rules prioritize all the jobs that are waiting for processing on a machine. Whenever a machine has been freed, a dispatching rule inspects the waiting jobs and selects the job with the highest priority. Some priority sequencing rules, e.g., First Come First Serve (FCFS), Earliest Due Date (EDD), Longest Processing Time (LPT) and Shortest Processing Time (SPT), are the common rules. These rules are called one-dimensional rules because they determine priority based on a single aspect of the job, such as arrival time at the workstation, the due date, or the processing time [23]. M. Arshad et al. [24] worked on the scheduling of different flexible manufacturing system (FMS) layouts using FCFS, SPT, LPT and EDD rules. For the four scheduling rules and three FMS layouts, simulation results were developed using Arena. The results showed that SPT rule performed better than the other scheduling rules. P.R. Philopoom et al. [25] examined the involvement of trade-offs while choosing a scheduling rule. The results showed that the SPT rule performed better for modest tardiness penalties. With the increase in penalty for tardiness, the FCFS rule worked well. Due to the interaction between the parameters of the due-date assignment rule and EDD rule, the EDD rule does not perform well. Using genetic algorithm and machine learning techniques, C.Y. Lee et al. [26] presented the job shop scheduler framework to schedule jobs. The proposed approach was compared with the conventional method and the results showed significant improvements. It was believed that the performance of the manufacturing systems will be impacted greatly after the successful implementation of the proposed integrated approach. To minimize the mean tardiness, M.X. Weng et al. [27] presented an efficient priority rule for scheduling job shops. The simulation results showed that the proposed rule outperformed the previous dispatching rules. N. Tyagi et al. [28] introduced five dispatching rules for single machine total tardiness scheduling problem to minimize the tardiness and the number of tardy jobs. Five dispatching rules were proposed for the problem and the performance of all dispatching rules were compared. It was found that EDD and SPT rules are better for the minimization of tardy jobs and makespan, respectively.

Production scheduling and control is a crucial component of Industry 4.0 initiatives because smart manufacturing relies on optimized production activities. Prior studies have mostly concentrated on the technological aspects of Industry 4.0, and little is known about how a Programmable Field Controller (PFC) is affected by digital capabilities and how it functions in this unique environment [29–37]. In this perspective, the major world powers have been compelled to develop their plans because of Industry 4.0 policy change. For instance, the United States started Smart Production, a fully integrated collaborative manufacturing system that responds in real time to meet changing conditions and demands in a factory [38,39]. To make society more sustainable and comfortable, Japan proposed Society 5.0 while Germany started Industry 4.0 concepts to digitize every aspect of society through smart planning and scheduling of manufactured products [40]. To diversify its

economy, grow public service sectors and lessen its reliance on oil, Saudi Arabia launched the Vision 2030 strategy framework in which smart production scheduling is one of the main pillars [41].

While working on the mathematical modeling for process optimization of the yogurt filling machine, Salah et al. [42] presented a model for filling yogurt and the required flavors at two separate points. The total length of the conveyor belt was divided into three equal parts. The different points on the conveyor belt were the entry point of the cups in the machine, the dairy product and flavor filling points, and the exit point of the cups from the machine. A set of orders received from customers was used to evaluate the model and the results were checked on the one-dimensional rules. A rule in which the prioritized performance measure resulted better than the other rules was selected for sequencing. While extending this work, Chen et al. [43] made slight modifications to the machine settings by dividing the conveyor belt into two equal parts and the filling of yogurt and flavors in the cups from a single point. The model resulted in a slightly reduced processing time.

In the current study, the machine settings are changed to dedicated filling point for each required flavor and a mathematical model is developed for the modified machine settings (Case-III). The outputs of the models of the previous similar studies, i.e., Salah et al.'s [42] model (Case-I) and Chen et al.'s [43] model (Case-II), are compared with the current model for the modified machine settings to find the one with better results than the other two. The results of the selected model (better than the other models) are used for sequencing the processing of orders using one-dimensional rules with the aim to find a rule which results better than the other ones for the most valued performance measure.

The article is organized as follows: a detailed introduction is provided in Section 1; the problem description is presented in Section 2; the modeling details are given in Section 3; Section 4 illustrates the solution procedure; sequencing the processing of orders is explained in Section 5; the results are discussed in Section 6; and Section 5 draws the important conclusions and future research directions.

## 2. Problem Description

The floor standing configuration of the dairy product filling machine is used for the filling of bottles with dairy product and flavors. Customers' demand may be a minimum of 0.25 L or a maximum of 1.5 L of dairy product mixed with a flavor. The demand can be for only the dairy product (Type-I), or sugar mixed with the dairy product (Type-II). For each type, separate containers in the machine are used. Three different flavors can be mixed with any type of dairy product, including flavor 1, flavor 2, and flavor 3. Customers' demands may be for only Type-I or Type-II dairy product or any type of dairy product mixed with any one of the three flavors. Table 1 shows all combinations of diverse types of dairy product and flavors that can be provided to the customers using the automatic filling machine.

**Table 1.** Different combinations of dairy product and flavors.

| Dairy Product (Type-I) | Dairy Product (Type-II) | Flavor 1 | Flavor 2 | Flavor 3 |
|:---:|:---:|:---:|:---:|:---:|
| | | 0 | 0 | 0 |
| | | 1 | 0 | 0 |
| 1 | 0 | 0 | 1 | 0 |
| | | 0 | 0 | 1 |
| | | 0 | 0 | 0 |
| | | 1 | 0 | 0 |
| 0 | 1 | 0 | 1 | 0 |
| | | 0 | 0 | 1 |

As illustrated in Figure 1, a large number of bottles are available for filling and these can be loaded on the conveyor belts. Bottles of required capacities are picked and placed on the conveyor belts which carry them towards the dedicated filling points. The binary numbers 1 and 0 show the motion and no-motion (stationary state) of the conveyor belts, respectively. There are two belts for each flavor and when one belt stops for filling of dairy product and a flavor, the other belt moves the empty or complete filled bottles from entry to filling or from filling to exit points, respectively. As an empty bottle reaches the filling point, it is filled with the required volumes of the dairy product and flavor, and the completely filled cup is then moved towards the exit point where a robotic arm can be used to pick and place the bottles in a tray.

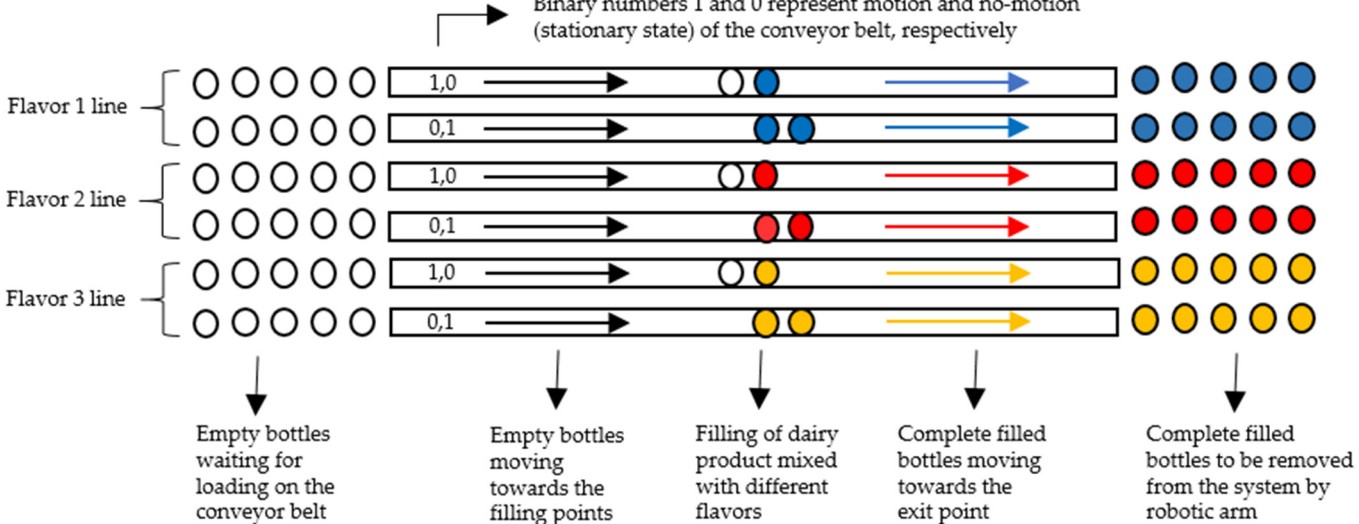

**Figure 1.** The processing of bottles of required volumes of yogurt and flavors with dedicated filling points for each flavor.

As shown in Figure 2, the solution to the problem is started by collecting all indices and parameters needed for the mathematical modeling. These indices and parameters are used in solving the models for Case-I, Case-II, and Case-III machine settings. Once the models are solved, the values of the objective functions of the three models are compared and the one with better results than the other two is selected for sequencing the processing of orders for all required flavors using the one-dimensional rules. After taking the average values of all performance measures, the one with better outcomes than the other two is chosen for order processing for all one-dimensional rules.

Considering the management specifications and machinery characteristics, two types of constraints are considered: operational and technological constraints. The operational constraints include the consideration of a range of total volume, satisfaction of customer demand within a given time, filling of a similar customer order in batches, and complete mixing of dairy product and flavors. The technological constraints include the limitations on the upper and lower volume of bottles; a limited number of containers, pumps, and nozzles for dairy product and flavors in the machine; feed rates of the solenoids valves for the dairy product and flavors; limitations on the number of dedicated lines for filling of flavors; limitation on the number of holders for bottles, a limited number of robotic arms for placement of empty bottles and removal of filled bottles; and limitation on the number of dairy product and flavor types.

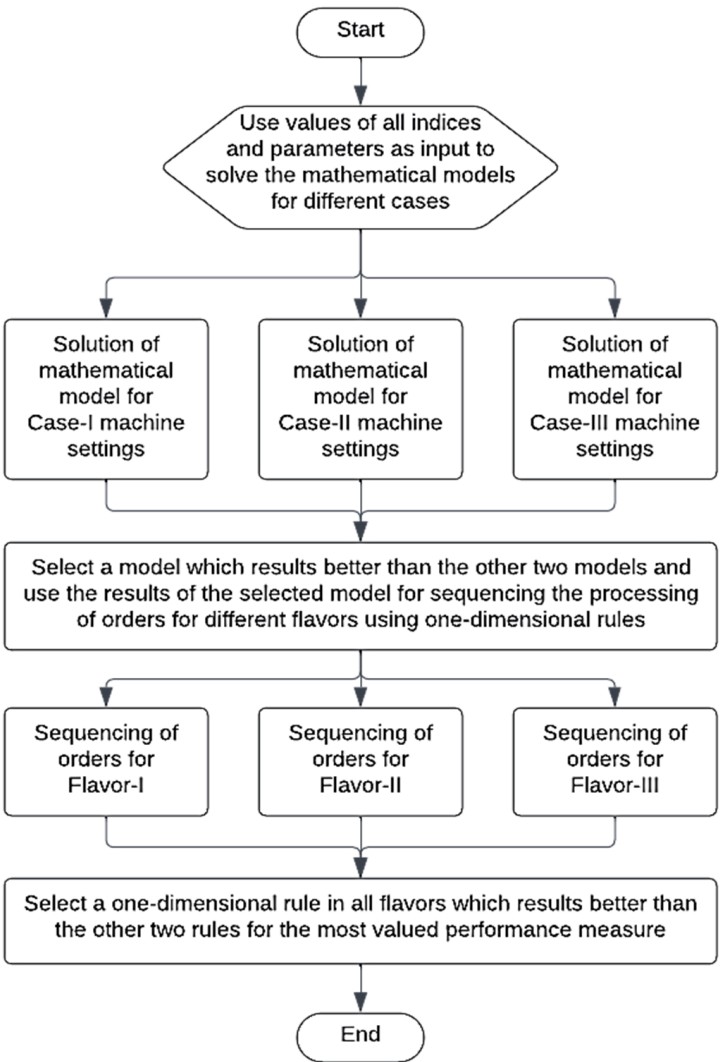

**Figure 2.** Optimal solution algorithm.

### 3. Mathematical Modelling

This presented research addresses the filling process of dairy product and various required flavors in bottles. In this model, the Type-I dairy product is provided to the customers and any of the three flavors are mixed with it. For each flavor, there is a dedicated filling point. The following assumptions are made in the problem formulation.

1. Dedicated filling point for each flavor.
2. A definite due date is assigned to each bottle.
3. Only one bottle is processed at any filling point at a time.
4. At the filling points, each bottle has a certain processing time.
5. The objective is to minimize the processing time of a set of orders.
6. No preemption is allowed once the filling of a set of orders is started.
7. All bottles and the filling points remain available for an unlimited period.
8. To process a bottle after an already-processed one, the setup time is considered zero.
9. The conveyor belt is divided into two equal parts and the three different points on the conveyor belt are entry, filling, and exit points.
10. The filling time of a bottle and its movement time between any two points on the conveyor belt are equal.

The indices, process parameters, and decision variables considered in the mathematical model include:

### Indices

| | | |
|---|---|---|
| i | dairy product percentage | $i = 1, 2, \ldots, I$ |
| y | dairy product type | $y = 1, 2, \ldots, Y$ |
| j | flavor percentage | $j = 1, 2, \ldots, J$ |
| f | flavor type | $f = 1, 2, \ldots, F$ |
| k | total volume | $k = 1, 2, \ldots, K$ |
| c | dairy product and flavor containers | $c = 1, 2, \ldots, C$ |
| n | number of bottles | $n = 1, 2, \ldots, N$ |

### Process Parameters

| | | |
|---|---|---|
| $S_b$ | speed of the conveyor belt | cm/s |
| $V_{iyjfk}$ | volume of dairy product | mL |
| $v_{iyjfk}$ | volume of flavor | mL |
| $W_{iyjfk}$ | waiting time for a demand | min |
| $V_{max}$ | maximum total volume of the bottles for dairy product and flavors | mL |
| $V_{min}$ | minimum total volume of the bottles for dairy product and flavors | mL |
| $V_c$ | volume of dairy product in a container | mL |
| $v_c$ | volume of flavor in a container | mL |
| $E_n$ | waiting time for the nth empty bottle at the entry point | s |
| $P$ | processing time of a bottle | s |
| $F_f$ | processing time of all orders for flavor of type f | s |
| $P_t$ | total processing time | min |
| $F_n$ | waiting time for the nth filled bottle at the exit point | s |
| $t_{iyjfk}$ | movement time from entry to filling point or filling to exit point | s |

### Decision variables

| | | |
|---|---|---|
| $\beta_{iyjfk}$ | dairy product valve feed rate | mL/s |
| $\gamma_{iyjfk}$ | flavor valve feed rate | mL/s |

For the convenience of the readers, few constraints and equations have been included from Chen et al. [43] in the current model. However, the objective function has been changed and model has been extended by including additional constraints and equations. This is also possible to write an integrated model; however, the constraints and equations of Chen et al. [43] are written separately as given below.

$$\frac{\beta_{iyjfk}}{V_{iyjfk}} l \leq Maximum \; S_b \; i = 1, 2, I \; y = 1, 2, \ldots, Y \; j = 1, 2, \ldots, J \; f = 1, 2, \ldots, F \; k = 1, 2, \ldots, K \tag{1}$$

$$\beta_{iyjfk} \leq Maximum \; \beta_{iyjfk} \; i = 1, 2, I \; y = 1, 2, \ldots, Y \; j = 1, 2, \ldots, J \; f = 1, 2, \ldots, F \; k = 1, 2, \ldots, K \tag{2}$$

$$E_n = (n-1)\frac{V_{iyjfk}}{\beta_{iyjfk}} \; i = 1, 2, \ldots, I \; y = 1, 2, \ldots, Y \; j = 1, 2, \ldots, J \; f = 1, 2, \ldots, F \; k = 1, 2, \ldots, K \tag{3}$$

$$P = 3\frac{V_{iyjfk}}{\beta_{iyjfk}} \; i = 1, 2, \ldots, I \; y = 1, 2, \ldots, Y \; j = 1, 2, \ldots, J \; f = 1, 2, \ldots, F \; k = 1, 2, \ldots, K \tag{4}$$

$$F_n = (n+2)\frac{V_{iyjfk}}{\beta_{iyjfk}} \; i = 1, 2, \ldots, I \; y = 1, 2, \ldots, Y \; j = 1, 2, \ldots, J \; f = 1, 2, \ldots, F \; k = 1, 2, \ldots, K \tag{5}$$

As already explained, the constraints (1) ensure that the speed of the conveyor belt must be less than or equal to its maximum allowable speed. Constraints (2) ensure that the feed rates of the dairy product controlled by solenoid valves must be less than or equal to

their maximum allowable feed rates. Equations (3)–(5) are used to find the waiting time of an empty cup at the entry point, the total processing time of a cup, and the waiting time to fill a cup at the exit point, respectively.

The new objective function minimizes the filling time which is linked with the feed rate of the solenoid valves and the needed dairy product volume. Mathematically, the objective function can be written in terms of the feed rate of solenoid valves and required volumes of dairy product and flavors as follows:

Minimize:

$$Z = \sum_{i=1}^{I} \sum_{y=1}^{Y} \sum_{j=1}^{J} \sum_{f=1}^{F} \sum_{k=1}^{K} \left( \frac{V_{iyjfk}}{\beta_{iyjfk}} + \frac{v_{iyjfk}}{\gamma_{iyjfk}} \right) \tag{6}$$

Subject to the constraints and equations as given below.

$$W_{iyjfk} \geq \frac{F_N}{60} \; i = 1, 2, \ldots, I \; y = 1, 2, \ldots, Y \; j = 1, 2, \ldots, J \; f = 1, 2, \ldots, F \; k = 1, 2, \ldots, K \tag{7}$$

$$t_{iyjfk} = \frac{V_{iyjfk}}{\beta_{iyjfk}} \; i = 1, 2, \ldots, I \; y = 1, 2, \ldots, Y \; j = 1, 2, \ldots, J \; f = 1, 2, \ldots, F \; k = 1, 2, \ldots, K \tag{8}$$

$$t_{iyjfk} = \frac{v_{iyjfk}}{\gamma_{iyjfk}} \; i = 1, 2, \ldots, I \; y = 1, 2, \ldots, Y \; j = 1, 2, \ldots, J \; f = 1, 2, \ldots, F \; k = 1, 2, \ldots, K \tag{9}$$

$$(V_{iyjfk} + v_{iyjfk}) \leq V_{max} \; i = 1, 2, I \; y = 1, 2, \ldots, Y \; j = 1, 2, \ldots, J \; f = 1, 2, \ldots, F \; k = 1, 2, \ldots, K \tag{10}$$

$$(V_{iyjfk} + v_{iyjfk}) \geq V_{min} \; i = 1, 2, \ldots I \; y = 1, 2, \ldots, Y \; j = 1, 2, \ldots, J \; f = 1, 2, \ldots, F \; k = 1, 2, \ldots, K \tag{11}$$

$$\sum_{i=1}^{I} \sum_{y=1}^{Y} \sum_{k=1}^{K} V_{iyk} \leq V_c \; c = 1, 2, \ldots C \tag{12}$$

$$\sum_{j=1}^{J} \sum_{f=1}^{F} \sum_{k=1}^{K} V_{jfk} \leq v_c \; c = 1, 2, \ldots C \tag{13}$$

$$A_n = \left( \frac{n+2}{n} \right) \frac{V_{iyjfk}}{\beta_{iyjfk}} \; i = 1, 2, \ldots, I \; y = 1, 2, \ldots, Y \; j = 1, 2, \ldots, J \; f = 1, 2, \ldots, F \; k = 1, 2, \ldots, K \tag{14}$$

$$P_t = Maximum \left( \frac{F_f}{60} \right) \; f = 1, 2, \ldots, F \tag{15}$$

The objective function (6) minimizes the filling time of the required volumes of the dairy product and flavors into a cup while the constraints (7) ensure that the customer waiting time for an order must be greater than or equal to the total processing time on the order. The constraints (8) and (9) express that the time in which a cup moves from entry to filling or filling to exit point must be equal to the filling time of the dairy product and a flavor into a cup, respectively. The constraints (10) and (11) satisfy that the total required volume of a dairy product and a flavor must be less than or equal to the maximum allowable total volume, and it should be greater than or equal to the minimum allowable total volume, respectively. Constraints (12) and (13) show that the total dairy product and flavors used in the filling of bottles must be less than or equal to the total volumes of the containers of dairy product and flavors, respectively. Equation (14) is used to find the filling time of an order while Equation (15) evaluates the maximum processing time of different orders for a single flavor.

## 4. Solution Procedure

Orders are accepted based on machine availability and considering the customer's waiting time. Customers may order up to a certain volume, and owing to machine limitations, any order that is below or beyond the lower or upper limits, respectively, is not

accepted. Any dairy product percentage of a bottle's total volume may be selected, and the flavor percentage is the difference between 100 and the dairy product percentage. Each order calls for a specific number of bottles with a specific delivery time.

The problem was solved using the PHP Simplex tool, which is available online for solving linear programming problems with no limitations on the number of decision variables, constraints and equations. PHP Simplex solved the problem using Two-Phase Simplex method on a Core i7 computer with 1.99 GHz processor and the results were obtained within a reasonable computational time. The tool can be accessed online on http://www.phpsimplex.com/en/.

There are twelve orders, each with a distinct total amount of dairy product and flavor, as given in Table 2. Only Type-I dairy product is demanded, and a container must have a far greater amount of dairy product volume than the required flavor volume. There is a need to quantify the number of bottles and the waiting times for each order considering the total volume.

**Table 2.** Orders for dairy product (Type-I) mixed with three different flavors and the corresponding waiting times.

| Order No. | Volume (mL) | Dairy Product (%) | F-I (%) | F-II (%) | F-III (%) | $D_{iyjfk}$ (Bottles) | $W_{iyjfk}$ (min) |
|---|---|---|---|---|---|---|---|
| 1 | 500 | 85 | 15 | 0 | 0 | 25 | 6 |
| 2 | 750 | 90 | 10 | 0 | 0 | 40 | 7 |
| 3 | 1000 | 90 | 10 | 0 | 0 | 30 | 5 |
| 4 | 1500 | 95 | 5 | 0 | 0 | 45 | 10 |
| 5 | 250 | 90 | 0 | 10 | 0 | 30 | 7 |
| 6 | 600 | 90 | 0 | 10 | 0 | 35 | 8 |
| 7 | 900 | 90 | 0 | 10 | 0 | 20 | 5 |
| 8 | 1200 | 85 | 0 | 15 | 0 | 30 | 9 |
| 9 | 560 | 95 | 0 | 0 | 5 | 40 | 8 |
| 10 | 800 | 95 | 0 | 0 | 5 | 35 | 10 |
| 11 | 840 | 95 | 0 | 0 | 5 | 25 | 5 |
| 12 | 1300 | 90 | 0 | 0 | 10 | 20 | 6 |

The values of all indices and customer-order-dependent parameters (dairy product volume, flavor volume, number of bottles, and waiting time) are given in Table 2, while the values of independent parameters are as follows:

| | |
|---|---|
| $S_b$ | 10 cm/s (maximum speed); |
| $L_t$ | 90 cm; |
| $V_{max}$ | 1500 mL; |
| $V_{min}$ | 250 mL; |
| $V_c$ | 300 L (Type-I dairy product); |
| $v_c$ | 15 L (each flavor); |
| $\beta_{iyjfk}$ | 150 mL/s (maximum feed rate of dairy product valve); |
| $\gamma_{iyjfk}$ | 50 mL/s (maximum feed rate of flavor valve). |

These indices and parameters are used as input to solve the mathematical models for Case-I, Case-II, and Case-III with a common objective to maximize the speed of the conveyor belt or minimize the processing time. For the same set of orders, the mathematical models for three cases are solved simultaneously to find the optimal values of the decision variables while satisfying all constraints and equations. The outcomes of the three models are presented in Table 3.

**Table 3.** Optimal solutions in three cases for the same set of customer orders.

| Case No. | Order No. | $\beta_{iyjfk}$ (mL/s) | $\gamma_{iyjfk}$ (mL/s) | $S_b$ (cm/s) | $E_N$ (s) | $P$ (s) | $F_N$ (s) |
|----------|-----------|------------------------|--------------------------|--------------|-----------|---------|-----------|
| I | 1 | 94.44 | 16.67 | 10.00 | 108.01 | 22.501 | 130.51 |
| | 2 | 150 | 16.67 | 10.00 | 175.50 | 22.500 | 198.00 |
| | 3 | 150 | 16.67 | 7.50 | 174.00 | 30.000 | 204.00 |
| | 4 | 150 | 7.89 | 4.74 | 418.00 | 47.500 | 465.50 |
| | 5 | 50 | 5.56 | 10.00 | 130.50 | 22.500 | 153.00 |
| | 6 | 120 | 13.33 | 10.00 | 153.00 | 22.500 | 175.50 |
| | 7 | 150 | 16.67 | 8.33 | 102.60 | 27.000 | 129.60 |
| | 8 | 150 | 26.47 | 6.62 | 197.20 | 34.000 | 231.20 |
| | 9 | 118.22 | 6.22 | 10.00 | 175.50 | 22.500 | 198.00 |
| | 10 | 150 | 7.89 | 8.88 | 172.27 | 25.333 | 197.60 |
| | 11 | 150 | 7.89 | 8.46 | 127.68 | 26.600 | 154.28 |
| | 12 | 150 | 16.67 | 5.77 | 148.20 | 39.000 | 187.20 |
| II | 1 | 94.44 | 16.67 | 10.00 | 108.01 | 13.501 | 121.51 |
| | 2 | 150 | 16.67 | 10.00 | 175.50 | 13.500 | 189.00 |
| | 3 | 150 | 16.67 | 7.50 | 174.00 | 18.000 | 192.00 |
| | 4 | 150 | 7.89 | 4.74 | 418.00 | 28.500 | 446.50 |
| | 5 | 50 | 5.56 | 10.00 | 130.50 | 13.500 | 144.00 |
| | 6 | 120 | 13.33 | 10.00 | 153.00 | 13.500 | 166.50 |
| | 7 | 150 | 16.67 | 8.33 | 102.60 | 16.200 | 118.80 |
| | 8 | 150 | 26.47 | 6.62 | 197.20 | 20.400 | 217.60 |
| | 9 | 118.22 | 6.22 | 10.00 | 175.50 | 13.500 | 189.00 |
| | 10 | 150 | 7.89 | 8.88 | 172.27 | 15.200 | 187.47 |
| | 11 | 150 | 7.89 | 8.46 | 127.68 | 15.960 | 143.64 |
| | 12 | 150 | 16.67 | 5.77 | 148.20 | 23.400 | 171.60 |
| III | 1 | 94.44 | 16.67 | 10.00 | 108.01 | 13.501 | 121.51 |
| | 2 | 150 | 16.67 | 10.00 | 175.50 | 13.500 | 189.00 |
| | 3 | 150 | 16.67 | 7.50 | 174.00 | 18.000 | 192.00 |
| | 4 | 150 | 7.89 | 4.74 | 418.00 | 28.500 | 446.50 |
| | 5 | 50 | 5.56 | 10.00 | 130.50 | 13.500 | 144.00 |
| | 6 | 120 | 13.33 | 10.00 | 153.00 | 13.500 | 166.50 |
| | 7 | 150 | 16.67 | 8.33 | 102.60 | 16.200 | 118.80 |
| | 8 | 150 | 26.47 | 6.62 | 197.20 | 20.400 | 217.60 |
| | 9 | 118.22 | 6.22 | 10.00 | 175.50 | 13.500 | 189.00 |
| | 10 | 150 | 7.89 | 8.88 | 172.27 | 15.200 | 187.47 |
| | 11 | 150 | 7.89 | 8.46 | 127.68 | 15.960 | 143.64 |
| | 12 | 150 | 16.67 | 5.77 | 148.20 | 23.400 | 171.60 |

It can be observed that the flow rate of nozzles is not increased once the maximum allowable speed limit of the conveyor belt is reached. Similarly, once the nozzles' maximum permissible flow rate is reached, the conveyor belt speed cannot be adjusted any further. The total volumes of dairy product and three different flavors needed to fulfill all the twelve orders are 271.92 L, 11.25 L, 10.05 L, and 6.17 L, respectively. The processing times in the

set of orders are less than the waiting times of customers for an order when calculated in three cases.

The processing times in Case-I and Case-II are 40.41 min and 38.13 min, respectively. While the processing time in Case-III is the maximum time while processing any of the flavors. In Case-III, the processing times of flavors 1, 2, and 3 are 15.82 min, 10.78 min, and 11.52 min, respectively, and the maximum time in three is 15.82 min. Considering these results, only Case-III is considered for sequencing the processing of orders using one-dimensional rules due the better outcomes than the other two models.

## 5. Sequencing the Processing of Orders Using One-Dimensional Rules

In Case-I and Case-II, the filling of dairy product mixed with different flavors cannot be performed simultaneously as the bottles are filled one after the other, and only the filling of the first order in the sequence is started at time zero. However, in Case-III, the filling of dairy product and different flavors in the bottles can be performed simultaneously as there are dedicated filling points for each flavor. Hence, the filling of dairy product mixed with any required flavor can be started at time zero.

There are twelve orders from clients for various dairy product flavors and volumes. Only the Type-I dairy product is mixed with three assorted flavors. The bottles are processed using EDD, FCFS and SPT rules for each demanded flavor of dairy product, and the necessary processing times are calculated. As given in Table 4, the orders for dairy product mixed with the three flavors have been processed through the machine using the three one-dimensional rules. It can be noted that orders 1–4, 5–8, and 9–12 are dairy product mixed with flavors 1, 2, and 3, respectively.

**Table 4.** The processing of bottles based on different one-dimensional rules.

| Flavor | One-Dimensional Rule | Order No. | Time since Order Arrived | Machine Starting Time | Processing Time | Finish Time | Flow Time | Due Time | Actual Pickup Time | Early Time | Time Past Due |
|---|---|---|---|---|---|---|---|---|---|---|---|
| Flavor 1 | EDD | 3 | 1 | 0 | 3.2 | 3.2 | 4.2 | 5 | 4.2 | 0.8 | 0 |
| | | 1 | 2 | 3.2 | 2.03 | 5.23 | 7.23 | 6 | 7.23 | 0 | 1.23 |
| | | 2 | 3 | 5.23 | 3.15 | 8.38 | 11.38 | 7 | 11.38 | 0 | 4.38 |
| | | 4 | 3 | 8.38 | 7.44 | 15.82 | 18.82 | 10 | 18.82 | 0 | 8.82 |
| | FCFS | 2 | 3 | 0 | 3.15 | 3.15 | 6.15 | 7 | 6.15 | 0.85 | 0 |
| | | 4 | 3 | 3.15 | 7.44 | 10.59 | 13.59 | 10 | 13.59 | 0 | 3.59 |
| | | 1 | 2 | 10.59 | 2.03 | 12.62 | 14.62 | 6 | 14.62 | 0 | 8.62 |
| | | 3 | 1 | 12.62 | 3.2 | 15.82 | 16.82 | 5 | 16.82 | 0 | 11.82 |
| | SPT | 1 | 2 | 0 | 2.03 | 2.03 | 4.03 | 6 | 4.03 | 1.97 | 0 |
| | | 2 | 3 | 2.03 | 3.15 | 5.18 | 8.18 | 7 | 8.18 | 0 | 1.18 |
| | | 3 | 1 | 5.18 | 3.2 | 8.38 | 9.38 | 5 | 9.38 | 0 | 4.38 |
| | | 4 | 3 | 8.38 | 7.44 | 15.82 | 18.82 | 10 | 18.82 | 0 | 8.82 |
| Flavor 2 | EDD | 7 | 1 | 0 | 1.98 | 1.98 | 2.98 | 5 | 2.98 | 2.02 | 0 |
| | | 5 | 3 | 1.98 | 2.4 | 4.38 | 7.38 | 7 | 7.38 | 0 | 0.38 |
| | | 6 | 2 | 4.38 | 2.78 | 7.16 | 9.16 | 8 | 9.16 | 0 | 1.16 |
| | | 8 | 4 | 7.16 | 3.63 | 10.79 | 14.79 | 9 | 14.79 | 0 | 5.79 |
| | FCFS | 8 | 4 | 0 | 3.63 | 3.63 | 7.63 | 9 | 7.63 | 1.37 | 0 |
| | | 5 | 3 | 3.63 | 2.4 | 6.03 | 9.03 | 7 | 9.03 | 0 | 2.03 |
| | | 6 | 2 | 6.03 | 2.78 | 8.81 | 10.81 | 8 | 10.81 | 0 | 2.81 |
| | | 7 | 1 | 8.81 | 1.98 | 10.79 | 11.79 | 5 | 11.79 | 0 | 6.79 |
| | SPT | 7 | 1 | 0 | 1.98 | 1.98 | 2.98 | 5 | 2.98 | 2.02 | 0 |
| | | 5 | 3 | 1.98 | 2.4 | 4.38 | 7.38 | 7 | 7.38 | 0 | 0.38 |
| | | 6 | 2 | 4.38 | 2.78 | 7.16 | 9.16 | 8 | 9.16 | 0 | 1.16 |
| | | 8 | 4 | 7.16 | 3.63 | 10.79 | 14.79 | 9 | 14.79 | 0 | 5.79 |

**Table 4.** *Cont.*

| Flavor | One-Dimensional Rule | Order No. | Time since Order Arrived | Machine Starting Time | Processing Time | Finish Time | Flow Time | Due Time | Actual Pickup Time | Early Time | Time Past Due |
|---|---|---|---|---|---|---|---|---|---|---|---|
| Flavor 3 | EDD | 11 | 1 | 0 | 2.39 | 2.39 | 3.39 | 5 | 3.39 | 1.61 | 0 |
| | | 12 | 2 | 2.39 | 2.86 | 5.25 | 7.25 | 6 | 7.25 | 0 | 1.25 |
| | | 9 | 2 | 5.25 | 3.15 | 8.4 | 10.4 | 8 | 10.4 | 0 | 2.4 |
| | | 10 | 3 | 8.4 | 3.12 | 11.52 | 14.52 | 10 | 14.52 | 0 | 4.52 |
| | FCFS | 10 | 3 | 0 | 3.12 | 3.12 | 6.12 | 10 | 6.12 | 3.88 | 0 |
| | | 9 | 2 | 3.12 | 3.15 | 6.27 | 8.27 | 8 | 8.27 | 0 | 0.27 |
| | | 12 | 2 | 6.27 | 2.86 | 9.13 | 11.13 | 6 | 11.13 | 0 | 5.13 |
| | | 11 | 1 | 9.13 | 2.39 | 11.52 | 12.52 | 5 | 12.52 | 0 | 7.52 |
| | SPT | 11 | 1 | 0 | 2.39 | 2.39 | 3.39 | 5 | 3.39 | 1.61 | 0 |
| | | 12 | 2 | 2.39 | 2.86 | 5.25 | 7.25 | 6 | 7.25 | 0 | 1.25 |
| | | 10 | 3 | 5.25 | 3.12 | 8.37 | 11.37 | 10 | 11.37 | 0 | 1.37 |
| | | 9 | 2 | 8.37 | 3.15 | 11.52 | 13.52 | 8 | 13.52 | 0 | 5.52 |

The order receiving time is the interval between the machine's starting time for the first order in the sequence until the bottle is filled, and the next order is queued. At time zero, the machine is turned on, and the filling of the first order in the sequence starts without any delay. The order has a certain processing time and once the bottle is filled with the required volumes of dairy product and flavors, the filling is stopped. The time at which the filling stops is the finish time. The flow time is the sum of the finish time of an order and the time since the order arrives. The actual pickup time is the time at which the order is ready and can be provided to a customer. Early time is the time when an order is received by the customer before the due time. The time past is the time when an order is received by the customer after the due time.

Where required, the unit of time is "minutes".

It can be observed that due to close due times, only one order is offered to customers before the due time. Out of four, the remaining three orders are provided after the due time, and one-dimensional rules are applied for the three flavors.

## 6. Results and Discussion

In Case-I, the filling of the dairy product and flavors is performed at two different points while the filling in Case-II takes place at a single point. Considering this, the processing time of Case-II is 1.06 times less than Case-I. The other reason that further reduces the processing time is the simultaneous filling of the required volumes of the dairy product and flavors at the dedicated filling points. As presented in Table 3, the processing times of each order in Case-II and Case-III are equal when orders in Case-III are not processed simultaneously. As shown in Figure 3, for non-simultaneous filling, Case-II and Case-III results are the same.

The total processing times of orders in Case-I and Case-II are 40.41 and 38.13 min, respectively. In Case-III, the processing times of orders are calculated by taking the maximum value of the sum of the processing times of orders for any flavor. To evaluate the processing times, Equation (15) can be used. The summation values of the processing times of flavors 1, 2, and 3 are 15.82, 10.78, and 11.53 min, respectively. In Case-III, when the filling of bottles takes place simultaneously at the dedicated filling points for each flavor, the total processing time of orders is reduced to 15.82 min. As shown in Figure 4, due to the simultaneous filling of assorted flavors mixed with dairy product at the dedicated filling points, the processing time in Case-III is 2.55 and 2.41 times faster than in Case-I and Case-II, respectively.

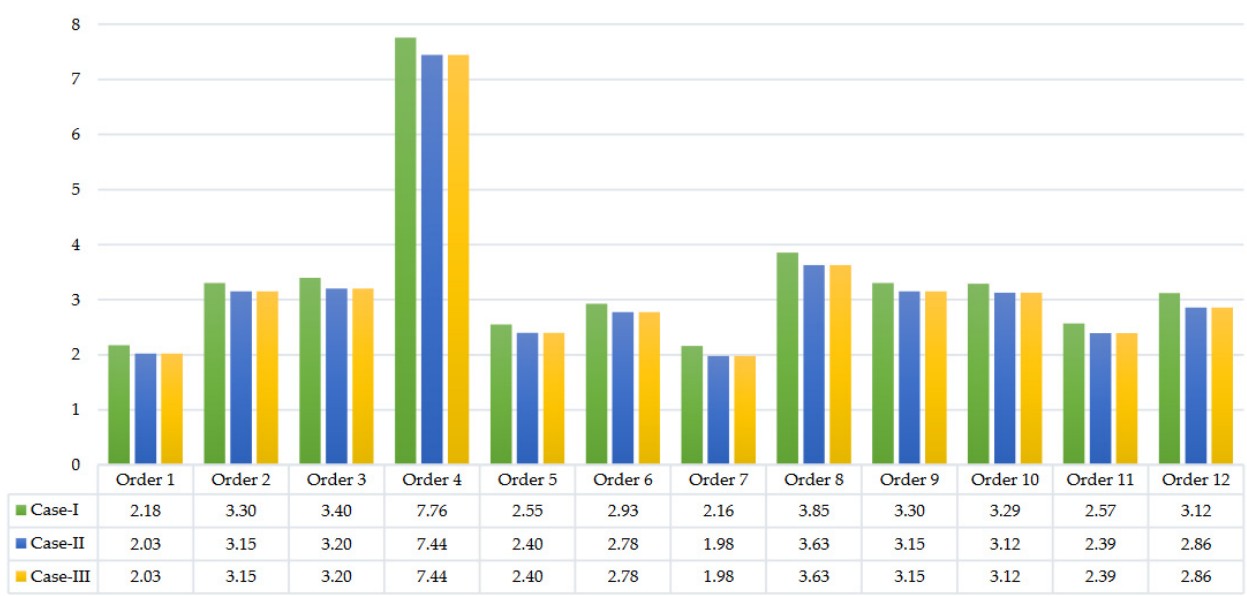

**Figure 3.** Processing time (minutes) of each order in different cases when the processing of orders in Case-III are considered non-simultaneously.

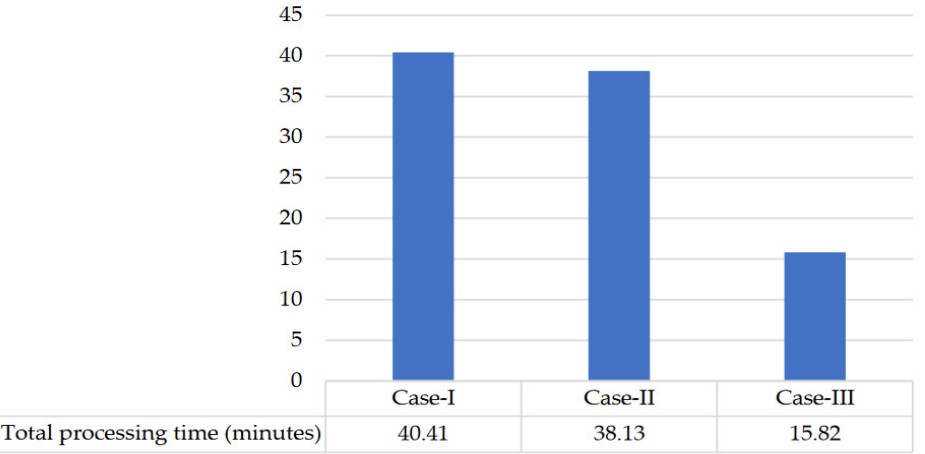

**Figure 4.** Total processing times (minutes) of all orders in different cases.

The three performance measures for managerial decisions include the average flow time, average time early, and average time past due. While meeting deadlines and keeping promises to customers, the maximum average early time, minimum average flow time, and average time past due are preferred. The values of all performance measures are calculated by using the three one-dimensional rules for each flavor, as shown in Figure 5. For the dairy product mixed with flavor 1, the SPT rules resulted in the maximum average flow time and average time past due. The same SPT rule also resulted in the maximum average early time. In the case of the dairy product mixed with flavor 2, the results of both the EDD and SPT were the same and better than the FCFS. Similarly, the EDD and SPT resulted in the same and better than the FCFS in the case of the dairy product mixed with flavor 3.

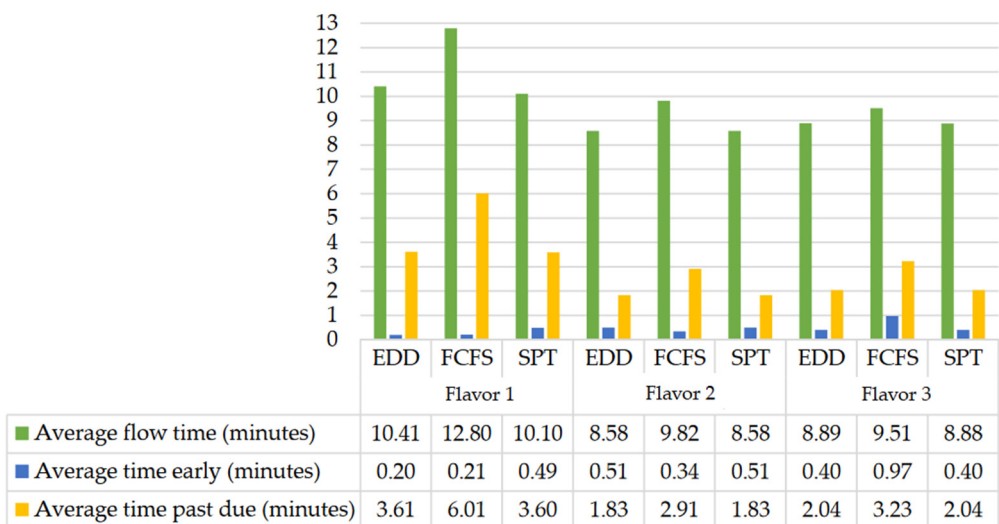

| | EDD | FCFS | SPT | EDD | FCFS | SPT | EDD | FCFS | SPT |
|---|---|---|---|---|---|---|---|---|---|
| | | Flavor 1 | | | Flavor 2 | | | Flavor 3 | |
| ■ Average flow time (minutes) | 10.41 | 12.80 | 10.10 | 8.58 | 9.82 | 8.58 | 8.89 | 9.51 | 8.88 |
| ■ Average time early (minutes) | 0.20 | 0.21 | 0.49 | 0.51 | 0.34 | 0.51 | 0.40 | 0.97 | 0.40 |
| ■ Average time past due (minutes) | 3.61 | 6.01 | 3.60 | 1.83 | 2.91 | 1.83 | 2.04 | 3.23 | 2.04 |

**Figure 5.** The values of performance measures for all flavors using three one-dimensional rules.

The average-time-past-due performance measure is valued the most by management, and the SPT rules produced better results than the other rules when dairy product was filled with the various required flavors. As a result, the bottles are handled following the SPT rule at the dedicated filling locations for each flavor.

## 7. Conclusions

In this research, the mathematical model for the modified machine settings produced better results than the models for previous machine settings. This research considered the earlier published models that were proposed for the same dairy product filling system. A mathematical model for the machine settings with a dedicated filling point for each flavor is developed and a real-life problem is solved.

The main objective was to minimize the processing time of filling the bottles with different required volumes of dairy product and flavors. The processing time was minimized by increasing the speed of the conveyor belt and tailoring the feed rate of dairy product valves considering their maximum allowable limits. When the machine setting in Case-I (filling of dairy product and flavors from two different points) was changed to Case-II (filling of dairy product and flavors at a single point), the processing time was marginal. However, when the machine setting was changed to Case-III (dedicated filling point for each flavor) and the filling of different flavors and dairy product was performed simultaneously, the results showed a significant reduction in processing time. Due to the optimal results, Case-III was selected for sequencing the processing of orders using the one-dimensional rules. Out of the three one-dimensional rules, the management preferred the SPT rules over the EDD and FCFS rules as SPT was better than the other rules for average time past due, which was the most valued performance measure for the management.

The limitations of the research include the mixing of only one type of flavor with the dairy product. In the future, the dairy product filling system can be made fully flexible to minimize or even eliminate the machine idle time at any dedicated filling point which would further reduce the processing time.

**Author Contributions:** Conceptualization, Y.C.; Methodology, Y.C. and X.Z.; Writing—original draft and presentation, Y.C., X.Z. and J.L.; Writing—review and editing, Y.C., X.Z. and J.L.; Funding acquisition, Y.C., X.Z. and J.L. All authors have read and agreed to the published version of the manuscript.

**Funding:** This work was supported by the Teaching Research and Reform Fund for SDJU and Shanghai multidirectional forging Engineering Technology Research Center (Grant No. 20DZ2253200), Aeronautical Science Foundation of China (Grant No. 2015ZB55002) and Natural Science Foundation of Henan Province (Grant No. 182300410239).

**Data Availability Statement:** The data presented in this study are available on request from the corresponding authors.

**Acknowledgments:** The authors acknowledge the support of Razaullah Khan for his comments to improve and refine this manuscript.

**Conflicts of Interest:** The authors report no conflict of interest.

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
