# Peer review of "Filling Process Optimization through Modifications in Machine Settings"

_processes, doi:10.3390/pr10112273_

Round 1

Reviewer 1 Report

the paper is introducing a new mathematical  model based on recent studies to improve the overall performance of a yogurt filling system using Industry 4.0 concepts.  The paper focused on improving the system parameters such as conveyor  belt speed, feed rates of solenoid valves, and waiting time of customers. The paper is well presented and the results demonstrate the main objectives of the paper.

However, some improvements can be made, such as :

1- PPC is not defined.

2- The authors must define what "one-dimensional rules" means for the readers and explain what distinguishes this research from other recent similar studies.

3- The authors must explain how the system receives the orders, or based on what the authors assumed the order list?

4- The authors mentioned  Industry 4.0, the authors "must" explain how the industry  4.0 concepts were represented in the mathematical model? 

5- The authors should explain how the mathematical model has been solved or which tool has been used to solve it?

Reviewer 2 Report

Review of the paper: processes-1928373:

 Process Optimization of an Automatic Yogurt Filling System by Modifying Machine Settings and Assessing Performance Measures for Managerial Decision

In this paper, the authors are addressing an optimization problem in order to determine the best parameters combination to enhance the performance of a Yougart filling system. In order to solve the obtained optimization problem, mathematical models are proposed and solved. In addition, several dispatching rules are used to deal with this problem as SPT and EDD. An experimental study based on real-life data is carried out to evaluate the performance of the proposed procedures.  This paper is presenting satisfactory scientific contributions and the obtained experimental results are providing strong evidence of the efficiency and the performance of the proposed procedures. For the above reasons, I recommend the acceptance of the paper.

Minor comments:

1.     Page 2, paragraph 2: Please provide a reference for the following claim “Most of the Saudi manufacturing sectors are relying on advanced technologies and smart production scheduling activities to meet the objective of Industry 4.0.”

2.     Page 5, in the “3. Mathematical Modellingsection,: replace “The following assumption is made in the problem formulation” with “The following assumptions are considered in the problem formulation”.

Reviewer 3 Report

·         Add a final statement in the abstract about the utility and applicability of the obtained results. Who will benefit? How will it be beneficial?

·         In page 6, along with the symbols and terminology of process parameters, the unit should also be mentioned.

·         In equation 1, ν is used, but it is not explained. What does it represent?

·         Acronyms like SPT, EDD, FCFS etc. should be expanded before first usage.

·    The future scope of the paper should be discussed in the final portion of the conclusion section.

·    The limitations of the work need to be looked into in the final portion of the conclusion section.

Round 2

Reviewer 1 Report

Dear author,

i have some concern regarding the originality.

Regards

Author Response

We are thankful to Reviewer 1 for his comments which improved our article significantly. Please note that responses to the comments of Reviewer 1 have been provided already and shared with the Assistant Editor.

Thank you.

Reviewer 3 Report

The paper can be accepted without any further changes.

Author Response

All authors of the article are thankful to the reviewer for his suggestions and encouraging comments. The required changes have been incorporated and the article has been updated accordingly.

Thank you.